DOI: 10.1038/s41467-018-04511-2　　**OPEN**

# Dynamics of self-reorganization explains passivation of silicate glasses

Stéphane Gin[1], Marie Collin[1], Patrick Jollivet[1], Maxime Fournier [1], Yves Minet[1], Laurent Dupuy[2], Thiruvilla Mahadevan[3], Sebastien Kerisit[4] & Jincheng Du [3]

Understanding the dissolution of silicate glasses and minerals from atomic to macroscopic levels is a challenge with major implications in geoscience and industry. One of the main uncertainties limiting the development of predictive models lies in the formation of an amorphous surface layer––called gel––that can in some circumstances control the reactivity of the buried interface. Here, we report experimental and simulation results deciphering the mechanisms by which the gel becomes passivating. The study conducted on a six-oxide borosilicate glass shows that gel reorganization involving high exchange rate of oxygen and low exchange rate of silicon is the key mechanism accounting for extremely low apparent water diffusivity ($\sim 10^{-21}\,\mathrm{m^2\,s^{-1}}$), which could be rate-limiting for the overall reaction. These findings could be used to improve kinetic models, and inspire the development of new molecular sieve materials with tailored properties as well as highly durable glass for application in extreme environments.

[1] CEA, DEN, DE2D, SEVT, 30207 Bagnols sur Cèze, France. [2] Tescan Analytics, ZAC St Charles, 13710 Fuveau, France. [3] Department of Materials Science and Engineering, University of North Texas, Denton, TX 76203, USA. [4] Pacific Northwest National Laboratory, Physical & Computational Sciences Directorate, Richland, WA 99352, USA. Correspondence and requests for materials should be addressed to S.G. (email: stephane.gin@cea.fr)

n nature, silicate glasses and minerals dissolve incongruently by various interrelated processes such as water diffusion in the solid, ion exchange, hydrolysis, condensation, and precipitation of secondary phases[1–6]. The global dissolution kinetics of the primary solid depends on its composition, its structure, and also on the fluid properties (e.g., pH, composition, and flow rate)[7]. As a result, the less-soluble elements will form new phases on the reacting surface, either by dissolution–reprecipitation[3,4,6] or by local rearrangement[8,9]. These phases can be amorphous or crystalline, dense, or porous[10]. These products change the solution composition; this can affect both the driving force for further dissolution and the mass transfer between the bulk solution and the reactive surface[1,6]. To date, the role of surface layers remains insufficiently understood to derive rate laws that can accurately predict the fate of silicate phases in various natural or engineered environments[10,11]. This is, however, an important need in many fields: in geochemistry to calculate the budget of some elements in the rivers and in the oceans, for the geological sequestration of $CO_2$, for the safety assessment of geological disposal of nuclear waste, etc. In the case of borosilicate glass used to confine nuclear wastes, several hypotheses are still under debate to account for the 3–5 orders-of-magnitude decrease of the dissolution rate measured in confined environments: (i) saturation of the solution in silica thereby decreasing the chemical driving force for the dissolution reaction[12], (ii) pore closure within the gel layer limiting the transport of aqueous species[9], (iii) water diffusion in the pristine solid[13], (iv) diffusion of dissolved silica through the gel layer leading to higher concentrations at the reaction front[2], and more recently (v) slow mobility of water molecules in the gel due to their confinement in constricted micropores[14,15]. The reasons why so many hypotheses are still under consideration are that high-resolution and high-sensitivity analytical tools available to investigate reacting buried interfaces are limited, and the composition and properties of alteration layers depend on many intrinsic and environmental parameters, complicating any attempt of a general theory[5,16].

Here, we propose an approach based on three experiments with isotopically tagged water molecules to study their mobility and their reactivity in a passivating gel formed on International Simple Glass (ISG), a six-oxide borosilicate reference glass[17], as well as molecular dynamics (MD) simulations and continuum-scale modeling to support experimental findings. Although isotopes have been used previously to investigate glass or mineral alteration mechanisms[6,18–21], another approach relying on the quantification of the passivating properties of the alteration layer is proposed here. The first experiment studies the dynamics of water in a gel formed during a 1-year-long alteration of ISG glass monoliths, and the second and the third experiments explore the dynamics of self-reorganization of a young and a mature gel, respectively. All the gels have been obtained by alteration of ISG at 90 °C, pH 7 in a solution initially saturated with respect to amorphous silica and containing K (called reference conditions in the following). It was demonstrated in a previous study that these conditions lead to the formation of a micrometer thick, uniform, and chemically homogeneous, passivating gel layer free of secondary phases[14]. Monoliths tested in experiments 1 and 3 come from the experiment described in Gin et al.[14], whereas experiment 2 is a short-term experiment conducted in similar conditions. From this study, we infer that hydrolysis–condensation reactions within the gel strongly affect the transport of water molecules and thereby limit glass dissolution.

## Results

**Water dynamics in a mature gel**. For the first experiment, we have selected a passivating gel formed by ISG glass alteration in the reference conditions. In this medium, after 1 year, the glass dissolution rate dropped by a factor ∼1000 relative to the initial dissolution rate and a 1.5-µm-thick gel-like material formed on

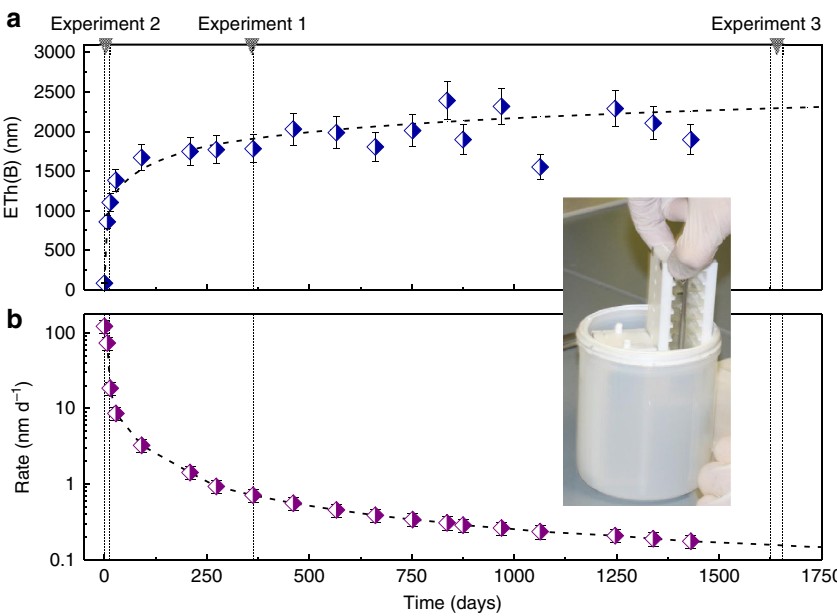

**Fig. 1** Experiment from which pieces studied in experiment 1 and 3 were taken. This initial experiment was conducted in the so-called "reference conditions" (90 °C, $pH_{90\ °C}$ 7, 160 mg $L^{-1}$ of Si and 6 g $L^{-1}$ of K). The image displays the reactor, the sample holders, and a few glass pieces. More details are provided in Gin et al.[14]. Experiment 2, although conducted on new fresh glass pieces corresponds to the early stages of the initial experiment. **a** Time evolution of the equivalent thickness of altered glass calculated from B concentration in solution. A few data points deviate from the overall trend because of the low concentration of B and the high dilution factor (∼11) prior to ICP-OES analyses. **b** Glass dissolution rate derived from the correlation displayed in **a**. Error bars represent s.d.

the glass surface (Fig. 1). Note that this value is more accurate than that estimated in a previous study[14] due to a larger number of data points giving a long-term trend. This film was partly characterized in a previous study[22] and its characterization was completed by the assessment of water content, speciation, and mean pore size[22,23] (Table 1 and Supplementary Note 1). This experiment therefore probes a uniform, 1.5-μm-thick amorphous gel formed by reorganization of the Al–Zr–silicate glassy network following the release of B, Na, and Ca and the incorporation of some K supplied by the leachate[14]. In this material, pore size is of the order of 1 nm. Six pieces of this altered glass were subsequently immersed at 25 °C in $H_2^{18}O$ ($^{18}O/^{16}O \sim 7$) for various durations up to 3 months and the oxygen isotopic profiles were recorded by time-of-flight secondary ion mass spectrometry (ToF-SIMS). Complementary analyses were carried out to validate the analytical protocol and verify the accuracy of the $^{18}O/^{16}O$ measurements (Supplementary Note 2).

Figure 2a shows that, for all contacting durations, $^{18}O$ exchanged with $^{16}O$ in the gel and penetrated up to the reactive interface, which was well defined by a sharp change in the B concentration (Fig. 2b). Moreover, the measurable increase of $^{18}O$ in the gel is observed even at the shortest contacting time (3 min) and its concentration significantly increases with time. We also note that all $^{18}O/^{16}O$ profiles are mostly flat with a slight but significant increase in the 200 nm near the reactive interface, likely due to the greater number of exchangeable O atoms in this less-mature area. According to the mean value of the O isotopic ratio measured in the gel, the time evolution of the percentage of exchanged O atoms was calculated assuming that only O from molecular water and silanols could exchange (Fig. 2c). It is found that only ~0.2% of these supposedly mobile O atoms had exchanged quickly (≤3 min). This fraction increases with time but remains below 6% even after 3 months (Table 2).

**A model for water diffusion in passivating gel.** The results from experiment 1 are consistent with a pore network made of a few deeply penetrating open channels and a large amount of dead-end or closed pores in which water molecules and silanols cannot exchange easily (Fig. 3). The basic assumption is that water molecules randomly explore the open porosity, and react with (Si, Al, and Zr)–O–Si bonds to open (by hydrolysis reactions) and close (by condensation reactions) pores within the gel. A model was developed according to these hypotheses (Supplementary Note 3). In this approach, the gel is assumed to contain spherical pores with a single mean pore radius. Some fraction of the pores are connected in aggregates that run from the solution/gel interface to the gel/pristine glass interface and are thus treated as cylinders for simplicity. The diffusion coefficient of water in connected pores (exchange mode 1), $D_c$, is derived from a fit to the O isotopic profile recorded at 3-min contacting time using the Crank solution of Fick's second law for a one-dimensional Cartesian case (Supplementary Fig. 4). Although the exact value cannot be determined, a value $\geq 10^{-11}\,m^2\,s^{-1}$ is expected to account for the flat experimental profile. Owing to the fast diffusion in connected pores, these pores rapidly equilibrate with the $^{18}O$-enriched solution and can serve as a continuous source of $^{18}O$ that diffuses through the gel, via gel reorganization, to reach closed pores. The diffusion coefficient for water diffusion out of the connected pores and into the gel (exchange mode 2), $D_d$, was calculated using an approximation to the Crank solution of Fick's second law in cylindrical geometry (see full derivation in Supplementary Information 3). This analysis led to values of $D_d$ that decreased with time following a power law (Supplementary Fig. 5), which could be indicative of a decrease in water diffusivity through the gel due to self-reorganization. $D_d$ decreased to $\sim 10^{-21}\,m^2\,s^{-1}$ within 24 h and reached $6 \times 10^{-23}\,m^2\,s^{-1}$ after 3 months. Because isotopic equilibration was not achieved even after 3 months, it is likely that water diffusion could slow down further with time.

To summarize, the gel behaves as a molecular sieve for water molecules with a few open pathways allowing rapid diffusion and many closed pores that reduce water diffusivity by at least eight orders of magnitude. This raises two questions: (i) do simulations at the pore scale support the idea that water diffusivity can be $\geq 10^{-11}\,m^2\,s^{-1}$ in open but narrow channels of the order of 1-nm width, and (ii) is there any experimental evidence that the gel undergoes hydrolysis and condensation reactions once it is formed, as was indirectly seen in mesoporous silica?[24]

In order to address the first question, water diffusion coefficients $D$ in amorphous silica micropores of diameter ranging from 0.5 to ~4 nm were evaluated by MD simulations with reactive potentials[25,26] (Supplementary Note 4). Pure silica was selected as reliable potentials have been developed for the silica-water system and as the actual gels are made of more than

### Table 1 Main features of the pristine and altered glass

| Parameter | Method | Result | Ref. |
|---|---|---|---|
| **Glass features** | | | |
| Nominal composition | – | $Si_{18.8}\,B_{9.6}\,Na_{7.6}\,Al_{2.3}\,Ca_{1.7}\,Zr_{0.5}\,O_{60.3}$ | [17] |
| Structure | NMR, MD simulations | Short-range and medium-range order data available in Collin et al. (2017) | [22] |
| Density (g cm$^3$) | Calculation + measurement | 2.5 | [22] |
| **Gel features** | | | |
| Dry composition | ICP-OES | $Si_{28.0}\,B_{0.3}\,K_{3.8}\,Al_{3.7}\,Ca_{0.5}\,Zr_{0.8}\,Na_{0.2}\,O_{62.7}$ | |
| Hydrated composition | ICP-OES, TGA, and NMR | $Si_{23.1}\,B_{0.3}\,K_{3.2}\,Al_{3.0}\,Ca_{0.4}\,Zr_{0.7}\,Na_{0.1}\,O_{51.8}\,H_{7.9}\,9.5\,(H_2O)$ | |
| Density (g cm$^3$) | Calculation | $2.25 \pm 0.10$ | [39] |
| Water content (wt%) | TGA | $12.5 \pm 0.1$ | |
| H distribution (%) | $^1H$ NMR | 70.7 $H_2O$, 23.5 H-bonded Si–OH, and 5.8 free Si–OH | |
| O distribution (%) | TGA, NMR | 71.6 $O_{BO}$, 12.9 $O_{NBO}$, and 15.5 $O_{H2O}$ | |
| Specific surface area (m$^2$ g$^{-1}$) | Calculation | $489 \pm 98$ | |
| Free volume (%) | TGA | $22 \pm 4$ | |
| Mean pore size diameter (nm) | Calculation | 1.1 | |

Details about the determination of H and O atoms distribution, specific surface area, free volume, and mean pore size are given in Supplementary Note 1. Pore size is too small to be experimentally determined. The value of ~1 nm is in good agreement with diffusion experiments with dyes[14]

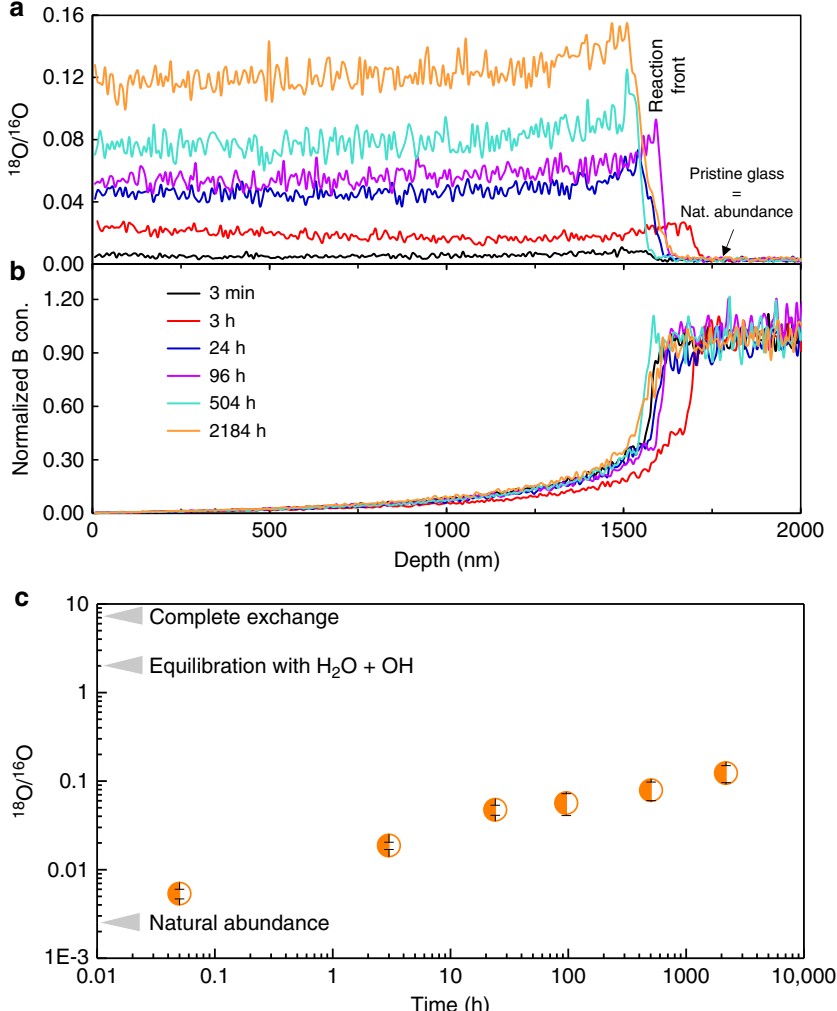

**Fig. 2** Water mobility in the gel. Water mobility is recorded by time-dependent isotopic and elemental ToF-SIMS profiles (experiment 1). For **a** and **b**, only lines are displayed for a better readability, and the analytical step is 7 nm. **a** Quantitative $^{18}O/^{16}O$ profiles in the gel for the different contacting durations. The mean value of the O isotopic ratio in the pristine glass ($2.2 \times 10^{-3}$) is within uncertainty of the natural abundance ($2.05 \times 10^{-3}$). **b** Normalized boron profiles for the same durations. The alteration front, given by the normalized concentration of 0.5 is located at 1574 ± 20 nm. The shifts between the different profiles give the analytical error of the technique, which is small. For a given contacting time, the B profile is well anti-correlated with the $^{18}O/^{16}O$ profile, indicating that some tagged water molecules reached the glass dissolution front at all tested durations. Some B is retained in the gel as evidenced by the B concentration on the gel side of the gel/pristine glass interface, which does not fall to zero. This likely corresponds to undissolved glass clusters[15]. **c** Evolution of the mean value of the oxygen isotopic ratio within the gel as a function of contacting time with the $H_2^{18}O$ solution. Error bars are calculated from uncertainites on natural abundance in the pristine glass

### Table 2 Fraction of exchanged mobile O

| Contacting time (h) | Fraction of $O_{H2O + -OH}$ exchanged |
|---|---|
| 0.05 | 0.002 |
| 3 | 0.009 |
| 24 | 0.023 |
| 96 | 0.027 |
| 504 | 0.038 |
| 2184 | 0.059 |

This fraction is based on the O speciation given in Table 1. The dilution of the isotopic ratio of the tracing solution by the pore water is negligible. Uncertainties are +/- 50%

90% of $SiO_2$ (Table 1). *D* values are plotted as a function of the radial distance from the center of the cylindrical pores (Fig. 4a). An unconstrained bulk diffusivity of $\sim 2.5 \times 10^{-9} \, m^2 \, s^{-1}$ [27,28] is attained only at the center of the largest pore. While similar results have been obtained with non-reactive potentials[29] with controlled addition of silanols, the use of reactive potentials in this work allows for dissociation of water, proton exchange, and formation of surface silanols. Close to the wall, water molecules form hydrogen bonds with oxygen in the silanols and other oxygen from the glass (Fig. 4a, b). The formation of hydrogen bonds of water with relatively immobile species in the glass leads to lowering of water diffusivity by an order of magnitude for water molecules that are closer to the pore wall. These slow water layers induce a drag on other water molecules and reduce diffusivity in the center, especially in smaller pores. If the pore diameter is large enough, water molecules at the center of the pore are hydrogen bonded only to the nearby mobile ones and hence have a diffusivity closer to that of water in bulk. These results show that, as long as the pores remain open and larger than 0.5 nm in diameter, water diffusion is not dramatically affected by interactions with pore walls. This important result

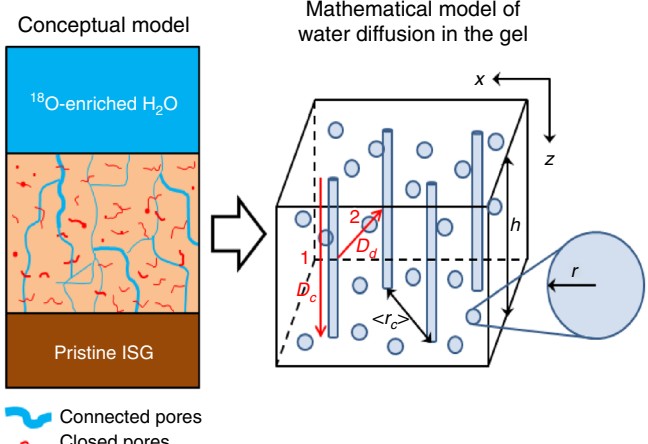

Conceptual model

Mathematical model of
water diffusion in the gel

**Fig. 3** A model for water diffusion in passivating gels. Conceptual model for interpreting water diffusion in passivating gel (left) and the corresponding mathematical model used to determine the water diffusion coefficients (right). The gel is assumed to contain spherical pores with mean pore radius, $r$, and a fraction of the pores are connected in aggregates that run from the solution/gel interface to the gel/pristine glass interface and are thus treated as cylinders with mean separation distance, $<r_c>$. Exchange modes 1 and 2 with diffusion coefficients $D_c$ and $D_d$, respectively, are illustrated in red

supports the idea that the value of $D_c$ derived from the modeling of the isotopic tracing experiments corresponds to water diffusion in open but constricted micropores, whereas $D_d$ is tied to diffusion between closed pores limited by bond breaking.

Furthermore, MD simulations were performed to obtain ISG glass and gel structures and analyzed the bottleneck and interstitial site density to understand available sites for water diffusion (Fig. 5 and Supplementary Note 4). Two gel structures were simulated from the glass. The first type of gel structure, called "Na-removed" in the following was obtained by removing Na atoms only from the ISG structure. The second type, chemically closer to the actual gel, and called "gel" in the following was prepared by removing all Na and B, 30% of Ca, and the free oxygen ions, i.e., oxygen ions that are not bonded to any other atom within a distance of 2.5 Å. The structures were generated using classical MD simulations with recently developed partial charge composition dependent potentials. A zoomed-in region of the 24,000 atoms simulation cell is shown in Fig. 5b. A Voronoi–Delaunay tetrahedron analysis[30,31] was performed to find the bottleneck size and interstitial site and density distributions. The results, which represent the available bottlenecks for water molecule diffusion, are shown in Fig. 5a. In the bulk glass structure, the density of bottlenecks with size larger than the size of a water molecule is relatively small. The density increases in the Na-removed structure. In the gel structure, the density of bottlenecks and bottlenecks with larger sizes are much higher, and both contribute to provide channels for water diffusion. The density of bottleneck size with 1.28-Å radius (about the size of a water molecule) is 1.2, 2.2, and 2.8/nm³ for ISG, Na-removed, and gel, respectively. These results are consistent with a recent MD study showing that a silica gel structure has randomly connected pores with certain pore size distributions that form channels for water diffusion[32]. Overall, this study at the nanometer scale suggests that the release of mobile species during glass alteration creates narrow channels in which water diffuses slightly slower than in the bulk, but not slow enough to account for the extremely low values derived from the

$^{18}$O-tracing experiments ($<10^{-21}$ m² s⁻¹). Moreover, the release of mobile species leads to a reorganization of the silicate network which might affect the diffusion pathways.

**Role of gel restructuring on passivation.** To address the second question, i.e., whether the gel undergoes hydrolysis and condensation reactions once it is formed, two experiments were conducted in the reference conditions, one focused on the early stages of gel formation (experiment 2), and the other focused on a mature gel (experiment 3). For experiment 2, two solutions were prepared: solution $S_1$ with natural water ($^{18}$O/$^{16}$O $= 2 \times 10^{-3}$) and solution $S_2$ with $^{18}$O-enriched water so that $^{18}$O/$^{16}$O $= 0.24$. Three polished pieces of unaltered ISG were first altered in solution $S_1$ for 3 days, after which one piece was withdrawn for ToF-SIMS depth-profiling analysis, while the other two were transferred to solution $S_2$ for an additional 4 or 10 days of alteration. Overall, O isotopic ratios were recorded in gels formed after 3, 7, and 13 days of alteration (Fig. 6). The gel thickness obtained from the B profiles was 500, 600, and 740 nm at day 3, 7, and 13, respectively (Fig. 6a). As expected, the gel analyzed after 3 days had the O isotopic signature of natural water. Importantly, no enrichment of $^{29}$Si was noticed in the gel (Fig. 6b, inset), confirming previous results indicating that, in silica-saturated and near-neutral pH conditions, Si from the glass does not dissolve, diffuse, and redeposit[14].

When in contact with solution $S_2$, the glass continued to alter but the isotopic signature of the gel originally formed in solution $S_1$ increased to a mean value of 0.15, which is very close to that recorded in the 100 nm of a new gel formed between days 3 and 7. From 7 to 13 days, the $^{18}$O/$^{16}$O ratio increased further in the part of the gel formed in natural water by an amount that is beyond the noise of the signal (0.15 → 0.17) as well as in the gel formed between day 3 and 7 (0.13 → 0.18). According to the O speciation in the gel (Table 1), and assuming that water molecules and silanols exchange first, a mass balance analysis indicates that 62% of the bridging O of the gel network was hydrolyzed and exchanged between days 3 and 7 and this figure increased to 70% at day 13. This is direct evidence that the young gel undergoes hydrolysis and condensation reactions that allow for isotopic exchange. Because Si atoms of the gel do not exchange with Si species in solution, it must be concluded that all the Si−O bonds of a given Si atom in the gel are not broken simultaneously. An alternative explanation, whereby all Si−O bonds are hydrolyzed, $H_4SiO_4$(aq) is released in pore water, and then quickly condenses has been ruled out in a previous study[14].

Experiment 3 explored the behavior of a mature gel. This gel resulted from the alteration of ISG that altered 1625 days in the reference conditions (Fig. 1). A piece of 1625 d was immersed for 1 month in a solution at 90 °C, pH 7, saturated with respect to amorphous silica, and spiked with $^{18}$O, so that $^{18}$O/$^{16}$O $= 0.24$. The O isotopic ratio was then recorded by ToF-SIMS depth profiling (Fig. 7). The shape of the O profile in the gel is similar to that obtained in tracing experiment 1 conducted at room temperature (Fig. 2a) but the $^{18}$O enrichment is much higher due to a greater reactivity at 90 °C than at room temperature[20]. In experiment 3, 21.1% of all O atoms available in the gel have been exchanged after 1 month. This value is close to the fraction of mobile O calculated from the budget of water molecules and silanols (28.4%, Table 1). However, experiment 1 demonstrated that a large faction of mobile O atoms are not easily accessible. To exchange these atoms between close and dead-end pores, network reorganization is necessary. This means that a total exchange of mobile O would necessarily be associated to some exchange of the O from the network, resulting in a higher $^{18}$O/$^{16}$O ratio than that at equilibrium. As a consequence, the mean $^{18}$O/$^{16}$O obtained

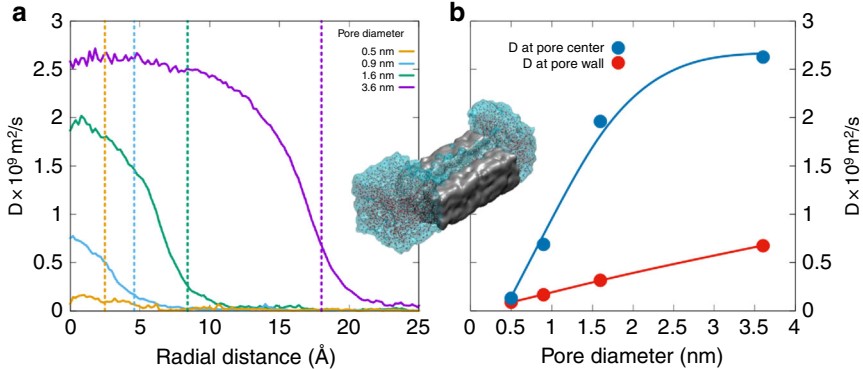

**Fig. 4** Water diffusivity in amorphous silica micropores. **a** Self-diffusion coefficient of water as a function of distance from the center of the pore in cylinder pores of amorphous silica with diameters of 0.5, 0.9, 1.6, and 3.6 nm from reactive force field-based molecular dynamics simulations. At the center of the pore, diffusing water molecules do not see the reactive sites in the wall and are relatively less constrained. The inner walls of the pore are oxygen terminated with atomic-scale roughness and reaction with water results in changes in the surface structure. Water molecules have been observed to diffuse up to a nanometer into the silica structure beyond the surface and further change the surface structure. The dashed lines corresponding to the color of the curves represent the approximate location of the pore wall. **b** Mean water diffusion coefficients at the pore wall and at the center of the pore for various pore sizes. In-between figures **a** and **b**, a snapshot of the 0.9-nm pore system with a section of the silica structure removed to show water in the pore. The silica structure is shown as the gray block and water on either side is shown as molecules inside the transparent blob

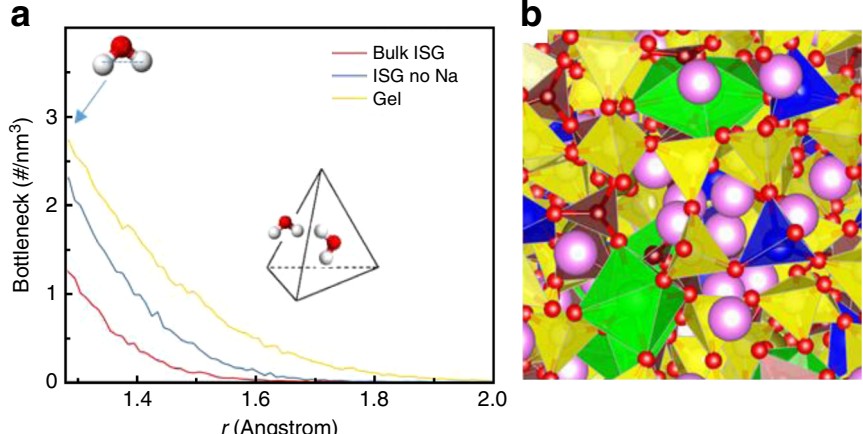

**Fig. 5** Bottleneck distribution in International Simple Glass and gel. **a** Bottleneck density distribution with bottlenecks larger than 1.28-Å radius for bulk ISG, Na-depleted ISG, and ISG gel structures. The inset shows a tetrahedron with a water molecule in the bottleneck (side face) and interstitial (center of the tetrahedron) sites. **b** Snapshot of a zoomed-in region of an ISG glass structure from MD simulations showing glass-forming network made up of $[SiO_4]$ (yellow), $[AlO_4]$ (blue), $[BO_4]$ and $[BO_3]$ (brown), and $[ZrO_6]$ (green) and charge compensator or network modifier elements, Na (pink) and Ca (golden). Oxygen atoms are displayed in red (see also Supplementary Fig. 6)

here $(5.3 \times 10^{-2})$ suggests that a fraction of mobile O and bridging O of the gel network were exchanged, although the exact amounts are unknown.

According to experiments 2 and 3, it is shown that both an early-formed and a mature gel could undergo hydrolysis and condensation reactions, suggesting that the gel reorganization is a continuous process affecting the pore connectivity and thus the mobility of water molecules. Although more research is needed to determine the kinetic constant and the activation energy of each type of reaction involving O, our results show that the consumption of water molecules in the exchange with silanols and bridging oxygens affects the flux of water molecules reaching the pristine glass surface. Indeed, a simple calculation based on the results of experiment 1 estimates that the flux of water molecules through the 1-year-old passivating gel is decreased by approximately four orders of magnitude compared to a non-passivating gel formed in a Si-free solution (Supplementary

Note 4). Previous studies proposed that hydrolysis, diffusion, and redeposition of Si were the key processes leading to the densification of the outer part of the gel, thereby closing open pores formed by the release of mobile species and eventually stopping further glass corrosion simply by preventing water molecules from reaching the pristine glass surface[9,33,34]. This paradigm must be revised as the present work shows that gel reorganization could be a continuous phenomenon involving a high reactivity of O and low mobility of Si. The simplistic idea that the gel is a rigid, strain-relaxed, and porous structure once formed, either totally inherited from the glassy network or re-precipitated, must also be revised, as its passivating properties mostly depend on reactions between water molecules and (Si, Al, and Zr)−O−Si bonds with bonding oxygen of the network. It is also expected that both the topology of the pore network and its dynamics of maturation depend on parameters such as glass composition, solution composition and pH, and temperature.

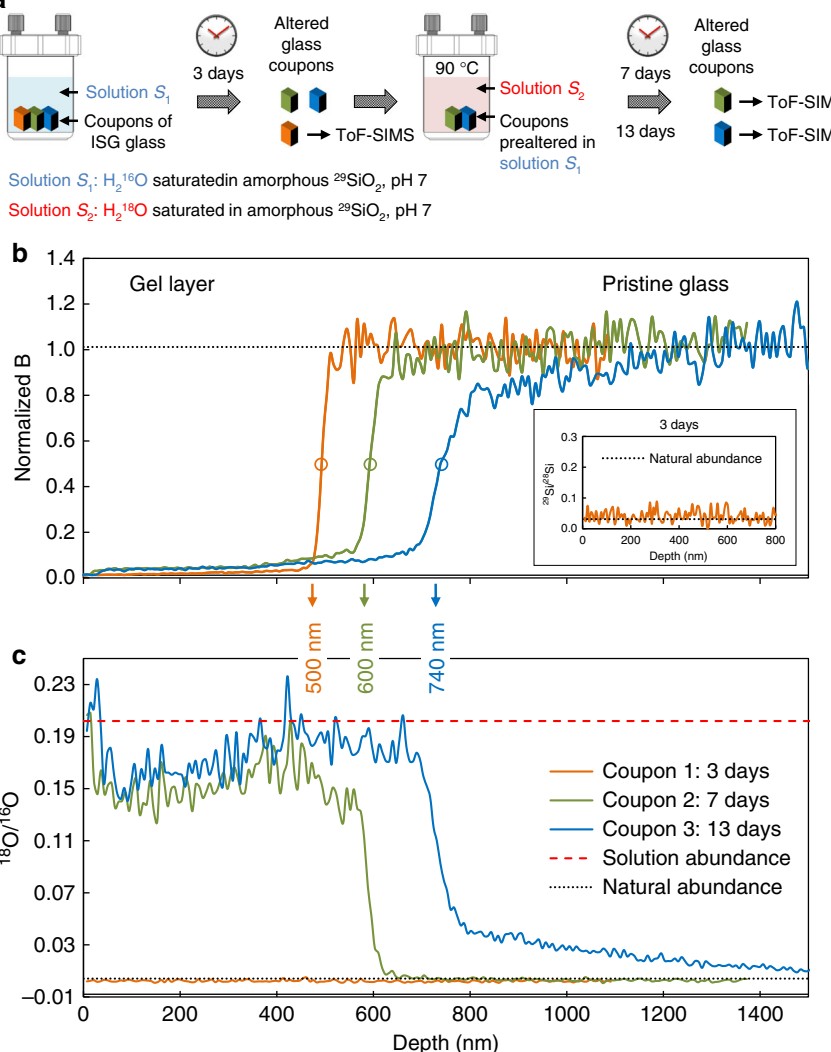

**Fig. 6** Dynamics of self-reorganization of a young gel. Dynamics is monitored by ToF-SIMS depth profiling (experiment 2). **a** Experimental protocol: piece 1 (orange) was altered for 3 days in solution $S_1$ only, piece 2 (green) was altered 3 days in solution $S_1$ followed by 4 days in solution $S_2$, and piece 3 (blue) is similar to piece 2 but the alteration duration in solution $S_2$ was 10 days. **b** Normalized B profiles showing the position of the alteration front at the three considered durations. The inset displays the $^{29}Si/^{28}Si$ ratio in the gel of piece 1. Although the glass is altered in a solution highly enriched in $^{29}Si$, the gel keeps the isotopic signature of the glass. **c** Oxygen isotopic profiles in the three gels. Contrary to Si, O in the gel displays an isotopic signature close to that of the solution. For ToF-SIMS profiles, only lines are displayed for a better readability, and the analytical step is 7 nm

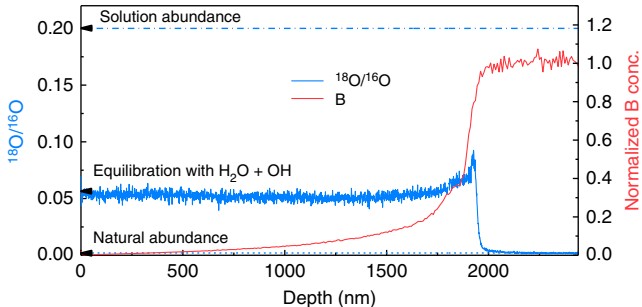

**Fig. 7** Dynamics of reorganization of a mature gel. Oxygen isotopic ratio and B profiles recorded by ToF-SIMS depth profiling on a piece altered 1625 days (experiment 3) in the initial experiment and then placed 1 month in similar conditions (90 °C, pH$_{90\,°C}$ 7, silica-saturated solution) with water spiked with $^{18}O$ ($^{18}O/^{16}O = 0.24$). During this 1-month experiment, the oxygen isotopic signature of the gel raised from natural abundance to 0.125 on average, leading to a fraction of exchanged O of 21.1%

Adjustment of these parameters––especially the relative proportion of low-strength bonds (e.g., Si−O−B) that might lead to rapid reorganization but also to extensive dissolution and high-strength bonds (e.g., Si−O−Zr)––could lead to the development of new materials, for instance, for controlled drug release or the fabrication of glasses resistant to extreme conditions. For nuclear glass, the design of a smart engineered barrier supplying the glass surface with elements rending the gel dense and chemically-and-radiation-resistant might contribute to improving the safety demonstration of geological repositories and eventually public acceptance.

## Methods

**Glass and experimental design.** ISG glass was prepared by MoSci Corporation (Rolla, MO, USA)[17]. Pieces were cut and polished to optical grade. The alteration experiment dedicated to the formation of the passivating gels studied in experiment 1 and 3 of the present work was conducted with 16 pieces placed in per-fluoroalkoxy (PFA) vessel, in controlled conditions: (90 ± 1) °C, pH$_{90\,°C}$ 7 ± 0.2, 6 g L$^{-1}$ K, and 160 ± 10 mg L$^{-1}$ of $^{29}Si$ corresponding to the saturation of amorphous silica. The reader is referred to Gin et al.[35] for more details about this experiment. In this initial experiment, glass alteration was monitored by solution sampling.

Samples were diluted and analyzed by inductively coupled plasma-optical emission spectroscopy (ICP-OES, Thermo Scientific ICAP 6300 DUO). Several pieces were withdrawn during the course of the initial experiment: one after 363 days for running experiment 1 of the present study and another after 1625 days for experiment 3.

For experiment 1, the 363-day piece withdrawn in the initial experiment was dried, broken into six pieces, and then placed at $(25 \pm 2)$ °C in a solution spiked with $H_2^{18}O$, so that $^{18}O/^{16}O = 7 \pm 3$. These pieces of altered glass were sampled after various durations and analyzed by ToF-SIMS.

Experiment 2 devoted to study the dynamics of the early stages of gel reorganization was performed with fresh ISG pieces at $(90 \pm 1)$ °C, $pH_{90°C}$ $7 \pm 0.2$, $3 \, g \, L^{-1}$ K using isotopically natural water in the first 3 days, and in $^{18}O$-enriched water between day 3 and 13, so that $^{18}O/^{16}O = 0.24 \pm 0.05$.

ToF-SIMS analyses were performed by Tescan Analytics, Fuveau, France using a ToF-SIMS5 spectrometer (IonTof––Münster, Germany). The analytical procedure was optimized to minimize exchange between pore water and air humidity and quantify oxygen isotopic ratio in the gel. Details are provided in Supplementary Note 2. A pulsed 25 keV ~0.1-pA $Bi_3^{++}$ primary ion source was employed over a rastered area of 50 μm × 50 μm (beam size ≤ 3 μm). Depth profiling was performed using a 2-keV $Cs^+$ sputter beam with a 45° incidence to the glass surface, giving a 195-nA target current over a 200-μm x 200-μm area. A flood gun was used to compensate the charging effect on the surface. Negative ion depth profiles were recorded. These conditions provide a step of 7 nm, similar in the gel and in the glass. Data acquisition and post-processing analyses were performed using the SurfaceLab 6 software. A profilometer was used to measure the crater depth at the end of the analysis. Data are displayed as a function of depth considering the same sputtering rate in the alteration layer and pristine glass.

**Molecular dynamics.** Molecular dynamics (MD) simulations of water diffusion in amorphous silica micropores were performed with reactive dissociable water potentials[25,26] and the LAMMPS package[36]. Reactive force field-based MD simulations were used previously to study the gel and gel/water interfaces[32,37]. The Hole program[38] was used to determine the location and smoothness of the pore walls. NVT simulations at 298 K were performed for all the systems: 8 ns for the 0.5 and 1-nm pore systems, 4 ns for the 2-nm pore system, and 2 ns for the 4-nm pore system. The diffusion coefficients were calculated as the average of the diffusion coefficient in the radial direction over the length of the pore. See Supplementary Note 4 for more details.

**Data availability.** The data that support the findings of this study are available from the corresponding author upon request.

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

## Acknowledgements

This work was supported as part of the Center for Performance and Design of Nuclear Waste Forms and Containers, an Energy Frontier Research Center funded by the U.S. Department of Energy, Office of Science, and Basic Energy Sciences under Award # DE-SC0016584. The authors are grateful to Géraldine Parisot for technical support.

## Author contributions

S.G. led the study and wrote the paper, M.C., P.J., and M.F. ran the experiments, L.D. conducted the ToF-SIMS analyses, T.M. and J.D. performed the MD simulations, and S. K. and Y.M. developed the diffusion model.

## Additional information

**Competing interests:** The authors declare that they have no competing interests.

