## [Peer Review File · Nature Communications]

Reviewer #2 (Remarks to the Author):

Understanding the dissolution of glasses, particularly in the context of nuclear waste vitrification and disposal, remains an important topic of investigation. This paper reports on the development of the gel layer on the surface of ISG. Although this paper does refer to the newly published paper by some of the same authors in *npj: Materials Degradation* (doi:10.1038/s41529-017-0025-y) in the context of characterisation of the gel layer I find it surprising that the discussion of the current manuscript does not refer back to this newly published paper.

One of the major conclusions of the paper is that gel reorganisation is a continuous process. This is not an entirely novel idea - for example the work by Cailleteau et al (*Nature Mat* doi:10.1038/nmat2301) involving some of the current authors pointed in the same direction ~a decade ago. To be fair new and different evidence in support of these ideas is presented here.

The use of different isotopes to investigate gel development is more novel, although other groups, most notably that of Geisler, have also utilised isotopically labelled solutions to try to increase understanding of glass dissolution. While the data as presented here show convincing profiles, I think it would be better if the actual datapoints were shown rather than just a line (even if the line is passing through the data points).

Based on the data from these experiments the gel layer on the surface of the glass is considered as a molecular sieve with a limited number of connected cylindrical pores and a greater number of closed pores. Diffusion through this gel is then modelled by using the complementary error function solution of Ficks second law applied to diffusion through the connected pores and separately to diffusion between disconnected or isolated pores. I do have some concerns that a solution that, mathematically at least assumes a continuum, is being applied to diffusion through nanometric scale pores which in reality will not be straight (at the very least is it possible to estimate their tortuosity to correct the length used in the simple model?), and is then used to infer a minimum value of the diffusion coefficient through these pores. MD, which does recognise the importance of the pore walls in this process, is used to support these values utilising 4 example pore sizes. Do you have any direct measurement of continuous pores of these dimensions in the actual gel layer?

The MD simulations are based on melt quenching of a cristobalite structure with some atoms removed. Thus it is only really looking at SiO₂. ISG contains B₂O₃ along with network modifiers. The fate of these species is necessarily involved in the development of the gel layer and the pore structure within it. The MD would be more convincing if at least it started with an alkali borosilicate structure.

More trivially the vertical columns in figure 2 could do with some labelling. Also why are they shown as multi-coloured bars?

I think that too much information in the paper has been relegated to the supplementary information, which is almost as long as the manuscript and which contains more figures (although figure S5 is simply a representation of a standard solution) than the manuscript. In particular one should not have to look at the supplementary information to discover that the complementary error function solution has been used to determine the diffusion coefficients.

Overall the experiments reported here are sound and the interpretation of them makes sense. However I am not completely convinced by the modelling used to support the interpretation and too many details of the modelling were relegated to the supplementary information.

Reviewer #3 (Remarks to the Author):

Upon review, I recommend the publication of this document. The authors may want to include language to answer to some of the comments below.

This is a novel study that provides key information on the current hypotheses regarding dissolution rate of glass. This work combines experimental and modeling tasks to understand water behavior in the gel.

This document is near-flawless; it provides key data and much-needed calculation details in the appendices.

I would only have general comments:

Does the term "condensation" include "precipitation" phenomena (a response to this comment should be included in the final document)? Could some of the pores be clogged by precipitates?

Is the pore clogging a dynamic phenomenon, or are some pores definitely closed while other are always open?

Finally, could a modeling exercise be implemented to understand the pH evolution of the pores? This might provide the further understanding of the solution chemistry occurring in the pores.

Answers to referees' comments

Article NCOMMS-17-34399-T “Dynamics of self-reorganization explains passivation of silicate glasses” by Gin et al.

Authors' answers are displayed in blue.

To address the general comments made by the editor and the reviewers we have improved the manuscript as follow:

- We have added a third experiment to reinforce our conclusions. This experiment performed on a mature gel provides the first evidence that the gel network evolves quickly through hydrolysis and condensation reactions although the glass corrodes almost 4 orders of magnitude slower than in the absence of a gel.
- We have better explained the diffusion model in the main text.
- We have added new MD simulations to reinforce our understanding of processes occurring at the pore level.
- We have extended the discussion and provided more details to improve the readability of the manuscript.

Reviewer #1:

I enjoyed reading the manuscript by Gin et al. Overall, this is a very nice piece of work on an important topic and I would recommend publication.

The major conclusion is that in nanoporous hydrated silica gel layers formed during glass corrosion, water penetrates deeply within the gel by molecular diffusion with a diffusion coefficient close to that observed in bulk liquid water (even through the average pore diameter is only ~1 nm), but that this rapid diffusion only accesses a relatively small fraction of the pore space. The rest of the pore space is accessed on a much longer time scale (~ 10^8 times more slowly) controlled by the rearrangement kinetics of the gel Si-O-Si network.

The authors' conclusions are strongly supported by their results and they significantly advance present understanding of glass corrosion, as far as I can tell. I have only a few minor comments:

1. Figure 2a in the manuscript is almost identical to Figure 7 in Bourg and Steefel, J. Phys. Chem. C 116:11556 (2012). The only difference is that the horizontal axis is reversed to represent distance from the pore center instead of distance from the pore walls. Everything else (from the choice of pore sizes to the actual reported values) is essentially identical. It would make sense to mention that the same results were obtained 5 years ago... This is a minor point since the MD simulations are not central to the manuscript, and it is nice to see that the reactive and non-reactive force fields for silica-water systems agree so nicely. *Good point, we have added this comment in the text. Note that here we wanted to use the most advanced MD potentials to study water diffusivity in pores even smaller than those investigated in the 2012 study.*

2. Would it be feasible to compare the time-scale of the slow water-exchange process to independent estimates of the time-scale of reorganization of the Si-O-Si network in water?

For example, experiments at pH 3.8 yield a rate constant of 0.03 min^{-1} for the condensation of SiO_2 monomers according to Rao & Gelb, J. Phys. Chem. B 108:12318 (2004), and references therein. Are similar estimates available near pH 7, and if so how do they compare with the measured kinetics of the slow process observed by the authors? This is a great idea, thank you. To do that we need to quantify separately O isotopic exchange on water molecules, -OH and bridging O. This has not been done in the present study. This is one potential extension of the present work. For that, one needs to use water spiked with ^{17}O and NMR spectroscopy in order to have a high species sensitivity. However it's not straightforward, and a new series of experiments must be performed to meet the constraints imposed by NMR (small particles of fully altered glass).

3. The work presented here has implications beyond the field of glass corrosion that may be worth mentioning, for example in studies of pressure dissolution and surface force measurements (e.g., Vigil et al., J. Colloid Interface Sci. 165:367 (1994), but there are likely more recent references). As suggested by other reviewers and because we have more room than initially expected, we have extended the discussion. We have read the above mentioned paper and haven't found other implications of our findings.

Reviewer #2:

Understanding the dissolution of glasses, particularly in the context of nuclear waste vitrification and disposal, remains an important topic of investigation. This paper reports on the development of the gel layer on the surface of ISG. Although this paper does refer to the newly published paper by some of the same authors in npj: Materials Degradation (doi:10.1038/s41529-017-0025-y) in the context of characterization of the gel layer I find it surprising that the discussion of the current manuscript does not refer back to this newly published paper. Collin et al., 2018 provided key quantities used in the present work to study and quantify the dynamics of water in gels. In particular, the quantification of O isotopic exchange was made possible thanks to the initial work on water speciation detailed in Collin et al., 2018. That paper was therefore cited when introducing the gel properties, but it did not touch upon the mechanisms by which the gel evolves with time and it was thus not directly included in the discussion.

One of the major conclusions of the paper is that gel reorganisation is a continuous process. This is not an entirely novel idea - for example the work by Cailleteau et al (Nature Mat doi:10.1038/nmat2301) involving some of the current authors pointed in the same direction ~a decade ago. To be fair new and different evidence in support of these ideas is presented here. Cailleteau and co-worker linked the sharp drop in the corrosion rate to the densification of the outer layers of the alteration film and explained that this densification is due to Si mobility and its ability to condense. This process leads to the clogging of the open porosity. The story told by Cailleteau ends once the porosity of the gel is closed since the glass seemed to no longer dissolve. We have learned from the O behavior that the story is more complicated. We also show that the gel can be protective even though Si is not mobile (i.e. does not dissolve completely before condensation). Moreover, we demonstrate that, although most of the porosity is closed, the gel is still very reactive and the flux of water reaching the pristine glass surface doesn't fall to zero. This could be of great interest for modeling the residual rate. We have extended the discussion in this direction.

The use of different isotopes to investigate gel development is more novel, although other groups, most notably that of Geisler, have also utilised isotopically labelled solutions to try to increase understanding of glass dissolution. Right, Geisler GCA 2015 is cited as it is an important paper in the field. We have added a few references showing that we are not the first group to use isotopes to decipher solid/liquid interactions.

While the data as presented here show convincing profiles, I think it would be better if the actual datapoints were shown rather than just a line (even if the line is passing through the data points). The readability is better using lines connecting data points rather than data points only. To inform the reader we have provided in the figure caption the analytical step.

Based on the data from these experiments the gel layer on the surface of the glass is considered as a molecular sieve with a limited number of connected cylindrical pores and a greater number of closed pores. Diffusion through this gel is then modelled by using the complementary error function solution of Ficks second law applied to diffusion through the connected pores and separately to diffusion between disconnected or isolated pores. I do have some concerns that a solution that, mathematically at least assumes a continuum, is being applied to diffusion through nanometric scale pores which in reality will not be straight (at the very least is it possible to estimate their tortuosity to correct the length used in the simple model?), and is then used to infer a minimum value of the diffusion coefficient through these pores. MD, which does recognise the importance of the pore walls in this process, is used to support these values utilising 4 example pore sizes. Do you have any direct measurement of continuous pores of these dimensions in the actual gel layer? The main text has been modified to better describe the diffusion model. We have refined the model by solving the Fick's equation in cylindrical geometry to account for water diffusion between closed pores. Nonetheless, we agree with the reviewer, the model is a simplified view of the real pore structure reflected in the experimental data. However, the model does allow one to derive a set of apparent diffusivities of water molecules in the gel. Further studies with certainly continue to improve the description the gel, as a dynamic materials (see for instance a comment (and the answer) by reviewer 1). To answer the last question, we don't have any direct evidence of long channels within the gel. Because the typical size is of the order of a nanometer, it is something extremely complicated to visualize. TEM observations are not successful as FIB prep are too thick. PNNL is developing a cryo FIB prep followed by a cryo APT analysis. It seems that this particular technique, which can keep water as it was in situ, might address this issue in the near future.

The MD simulations are based on melt quenching of a cristobalite structure with some atoms removed. Thus it is only really looking at SiO₂. ISG contains B₂O₃ along with network modifiers. The fate of these species is necessarily involved in the development of the gel layer and the pore structure within it. The MD would be more convincing if at least it started with an alkali borosilicate structure. We have better justified in the text the use of silica nanopores as a model system to study water diffusivity in constricted geometries. We have also added simulations of gel structures to investigate bottlenecks and interstitials in which water molecules can diffuse.

More trivially the vertical columns in figure 2 could do with some labelling. Also why are they shown as multi-coloured bars? We have made some changes to improve the clarity of the figure.

I think that too much information in the paper has been relegated to the supplementary information, which is almost as long as the manuscript and which contains more figures (although figure S5 is simply a representation of a standard solution) than the manuscript. In particular one should not have to look at the supplementary information to discover that the complementary error function solution has been used to determine the diffusion coefficients. We have added a significant amount of information to the main text in the revised version of the paper, including with respect to the model used to derive the water diffusion coefficients.

Overall the experiments reported here are sound and the interpretation of them makes sense. However I am not completely convinced by the modelling used to support the interpretation and too many details of the modelling were relegated to the supplementary

information.

Reviewer #3:

Upon review, I recommend the publication of this document. The authors may want to include language to answer to some of the comments below.

This is a novel study that provides key information on the current hypotheses regarding dissolution rate of glass. This work combines experimental and modeling tasks to understand water behavior in the gel.

This document is near-flawless; it provides key data and much-needed calculation details in the appendices.

I would only have general comments.

- Does the term "condensation" include "precipitation" phenomena (a response to this comment should be included in the final document)? Could some of the pores be clogged by precipitates? In fact no, if one consider that precipitation reactions take place from solution (bulk or pore). It is shown in this study (experiment 2) and in previous ones (*Gin et al., Nature Communications* 2015 and *Gin et al., Geochimica et Cosmochimica Acta* 2017) that the transformation of the glass into gel occurs without complete dissolution of glass formers. So precipitation of SiO_2aq in pores can be ruled out. It means that in situ hydrolysis condensation with a low mobility of glass formers dominates the gel formation and its evolution in the conditions of the present study.
- Is the pore clogging a dynamic phenomenon, or are some pores definitely closed while other are always open? Unfortunately, it's not yet possible to be conclusive, however experiment 2 reported in this paper clearly shows that the gel, once formed, continues to experience hydrolysis and condensation reactions without dissolution of glass formers (Si, Al and Zr in our case). It is thus likely that because of this fast dynamics pores open and close. A study performed in collaboration by Penn State University (not yet published) shows that a very mature gel obtained after 1625 days in the same conditions as those used in our study displays pores of approximately 2 nm whereas younger gels have much smaller pores. It would suggest that the reorganization never stops and that the pore network slowly evolves even though the external conditions remain the same. Further work will focus on this dynamics.
- Finally, could a modeling exercise be implemented to understand the pH evolution of the pores? This might provide the further understanding of the solution chemistry occurring in the pores. At the scale of a nanopore, local concentrations, cations hydration, and water speciation need to be calculated by MD with a dissociable reactive potential or from first principles. As water molecules are allowed to dissociate, proton transfer and water/glass reaction can happen. In this work, the MD simulations were performed in silica pores and at neutral conditions. Future work will be performed in alkali containing glasses and solutions where the pH value can evolve in reactive force based simulations.

Reviewer #1 (Remarks to the Author):

The authors' response to all review comments is satisfactory. The additional text, figures, and references significantly enhance the manuscript. Overall, the research looks excellent and the conclusions are well supported.

My only comments regard the text on page 13: on line 237, I would replace "demonstrates" with "suggests"; on line 234, what is meaningful about the value of 1.28 Å (wouldn't, e.g., the radius of water be more meaningful)?

Response to the reviewers' comments on the manuscript

Manuscript: "Dynamics of self-reorganization explains passivation of silicate glasses"

Authors: S. Gin, M. Collin, P. Jollivet, M. Fournier, Y. Minet, L. Dupuy, T. Mahadevan, S. Kerisit, J. Du

Reviewer #1

The authors' response to all review comments is satisfactory. The additional text, figures, and references significantly enhance the manuscript. Overall, the research looks excellent and the conclusions are well supported.

My only comments regard the text on page 13:

on line 237, I would replace "demonstrates" with "suggests"

Done.

on line 234, what is meaningful about the value of 1.28 Å (wouldn't, e.g., the radius of water be more meaningful)?

The radius of 1.28 Å is about half of the size of a water molecule which is around 2.75 Å (as the measure of interstitial sites is in radius so the diameter is about that of a water molecule). 1.28 Å is also the radius of helium atom, which makes water molecule one of the smallest molecules. This number was used in Mansas *et al. J. Phys. Chem. C* 121 (2017) 16201 (Supplementary Figure 6) in discussion water diffusion in model borosilicate glasses.

The sentence becomes: "The density of bottleneck size with 1.28 Å radius (about the size of a water molecule) is ..."

Reviewer #2

I am pleased that the bulk of my comments have been met in the paper. I do note that Fig 3 in the new manuscript and Fig S4 in the supplementary material are identical - I think the latter can be removed. Apart from this minor issue I am now happy to recommend publication.

Done.

Reviewer #3

Chose to give comments to the editor only, but supports publication.